# Early Alcohol Use Initiation, Obesity, Not Breastfeeding, and Residence in a Rural Area as Risk Factors for Breast Cancer: A Case-Control Study

**DOI:** 10.3390/cancers13163925

**Published:** 2021-08-04

**Authors:** Dorota Anna Dydjow-Bendek, Paweł Zagożdżon

**Affiliations:** Department of Hygiene and Epidemiology, Medical University of Gdansk, 80-211 Gdansk, Poland; pzagoz@gumed.edu.pl

**Keywords:** breast cancer, alcohol use initiation, BMI, rural area, breastfeeding

## Abstract

**Simple Summary:**

Breast cancer became the most common cancer globally in 2021, according to the World Health Organization. The aim of the study was to evaluate risk factors for breast cancer, such as early alcohol use initiation, obesity, breastfeeding, and place of residence. The effect of alcohol consumption by girls has been assessed in only a few studies and is not fully understood. In this study, it has been found to be associated with a higher risk of breast cancer. Our study also shed light on the incidence disparity—women were more at risk in the countryside than in the city. The results of this study should be included in the preparation of breast cancer prevention programs and also aimed at women in adolescence and early adulthood because exposures during childhood and adolescence can affect a woman’s long-term risk of breast cancer. Every effort should also be made to ensure that access to knowledge is open to all, regardless of where they live, giving all women equal opportunities.

**Abstract:**

The aim of this study was to determine the risk factors for breast cancer in the Polish population. In total, 201 Polish women newly diagnosed with breast cancer and 201 one-to-one age-matched healthy controls participated in this case-control study. Data on sociodemographic characteristics, reproductive and menstrual history, medical history, lifestyle factors, and anthropometric measurements were collected by the interviewers. Odds ratios and 95% confidence intervals were obtained using multivariate unconditional logistic regression models controlling for potential confounders. Significant relationships were observed between BMI, alcohol use initiation, breastfeeding, education, and place of residence. Obese women had a higher risk of breast cancer than women with a BMI < 30 (OR = 1.9; 95% CI: 1.16 to 3.04). Early alcohol use initiation (≤15 years) was associated with an almost two-fold higher risk of breast cancer (OR = 1.98, 95% CI: 1.06 to 3.69). Breastfeeding for less than 3 months (OR = 2.3, 95% CI: 1.52 to 3.5), receiving a basic and vocational education (OR = 2.5, 95% CI: 1.49 to 4.19), and living in a rural area (OR = 1.7, 95% CI: 1.05 to 2.9) increased the risk of breast cancer. Prevention activities for breast cancer are already needed in adolescents and young women. A much greater emphasis should also be placed on breast cancer prevention campaigns in rural areas in Poland.

## 1. Introduction

With more than 1.5 million new cases diagnosed each year, breast cancer is the most common cancer in the world and the leading cause of cancer-related deaths among women. This number continues to grow, especially in highly developed countries [1,2,3,4]. The risk factors for breast cancer include a positive family history, early menstruation, late menopause, late age at first pregnancy, length of breastfeeding period, use of hormone replacement therapy (HRT), and Caucasian race [5,6,7].

Because most of the risk factors for breast cancer are unmodifiable, the emphasis should be placed on those that can be modified, such as early alcohol consumption. Although it may seem that this factor has been studied extensively, its impact on women’s health requires more detailed analyses. While alcohol consumption among adult women has been consistently associated with an increased risk of breast cancer [7,8], little is known about the effects of alcohol consumption in early life when the mammary glandular tissue is particularly sensitive [9]. One study has suggested that early alcohol consumption may increase the relative risk of breast cancer even beyond menopause [10]; however, other research projects have indicated that higher alcohol consumption was associated with an increased risk of breast cancer only among women with both a family history of breast cancer and lower folate intake, not with the risk of breast cancer overall [11]. Therefore, a clearer understanding of the effects of exposures in early life is warranted, especially early alcohol consumption among women who are not aware of the effects of alcohol on the risk of breast cancer [12]. Data from the Central Statistical Office in Poland indicate a significant increase in alcohol consumption in Poland in the 1960s and 1970s. During this period, the study participants were teenagers, and an increase in the number of girls who became inebriated was noted [13]. Currently, scientific evidence on early alcohol consumption as a risk factor for breast cancer is lacking; therefore, it is necessary to conduct extensive studies aimed at filling this gap, which may turn out to be a key element of prevention.

Another risk factor discussed in the current literature is breastfeeding, which has a protective effect on the mother later in life [8]; nonetheless, the effects of breastfeeding on breast cancer requires additional research. Some studies have suggested that breastfeeding may act as a risk factor for breast cancer [14]. Despite this, breastfeeding should be recommended (due to its protective effects against BRCA1 mutation) [15]. Unfortunately, according to the WHO, the European region has the lowest breastfeeding rates of all regions in the world. Between 2006 and 2012, only 25% of infants were exclusively breastfed in the first 6 months of life [16].

This study also discussed other important factors that increase the risk of breast cancer, especially obesity in the post-menopausal period [17,18,19,20]. The significance of other risk factors, such as living in a rural area, is not well understood. Some hypotheses indicate that the potential connection between living in a rural area and breast cancer may be related to exposure to organochlorine pesticides [21]. Another explanation could be the high alcohol consumption in Poland after World War II, especially in the countryside [22]. Moreover, the fact that obesity is more common among women in rural areas may contribute to the higher incidence of breast cancer in these areas [23,24].

It is estimated that more than one-fourth of breast cancer cases could be prevented by maintaining a healthy body weight, abstaining from alcohol, engaging in physical activity, and breastfeeding, thereby reducing the exposure of women to exogenous oestrogens [25].

Examining the risk factors for breast cancer in Polish women may expand the current knowledge on this important topic and act as one of the building blocks of prevention. To our knowledge, alcohol use initiation in the context of breast cancer has not been analysed in the Polish population thus far. Therefore, the aim of our study was to assess the relationship between breast cancer risk and early alcohol consumption, place of residence (city/village), breastfeeding, and obesity in a sample of Polish women.

## 2. Materials and Methods

This study was designed as a case-control study with face-to-face interviews. The participants were selected from among Polish women diagnosed with incident, primary, and histologically-confirmed breast cancer (*n* = 201) who were admitted to the Oncological Surgery Clinic of the Medical University of Gdansk between January 2015 and June 2017, with diagnoses of breast cancer made no more than three months prior to the study. Patients diagnosed more than three months in advance of the study were not included, with the aim of controlling for changes in dietary habits or other behaviours. Additional inclusion criteria for cases were as follows: age between 50 and 69 years, provided consent to participate, and the ability to understand and answer the questionnaire. Women previously diagnosed with breast cancer and women who were unable to answer the questionnaire due to health, language, or educational barriers were excluded. During the same period, breast cancer (BCA) patients received an additional questionnaire to give to healthy women of the same age in the same place of residence who were of a similar economic status. A total of 201 female controls without any clinical symptoms or other signs or suspicion of any type of cancer in their medical history were selected. The questionnaire included a question about the date and result of the last mammogram; only women with normal mammography outcomes were eligible to be enrolled as controls. The participants in the control group were matched in terms of age (≥50) with the cancer patients and selected among residents of the same areas as the cases. Interviewers visited the participants who were selected as controls or asked the participants to complete the questionnaire and return it by mail. If a suitable control could not be selected among women in the BCA community, the researchers attempted to find an appropriate control using their own social network (20% of cases).

### 2.1. Sample Size Calculation

A sample size of 402 women (201 cases and 201 controls) was estimated to be necessary to observe a significant association between early alcohol use initiation and the risk of breast cancer. The sample size calculation was performed using the standard formula [26], taking into consideration an estimated prevalence of early alcohol use initiation among controls of 29% [27], an odds ratio of 1.8, a case-to-control ratio of 1:1, a power of 80%, and a significance level of 5%. The sample size analysis was performed in Stata (StataCorp. 2013. Stata Statistical Software: Release 15.1.; StataCorp LP, College Station, TX, USA) using commands that accounted for the matched case-control nature of the study design.

### 2.2. Lifestyle Assessment

The self-created questionnaire consisted of questions regarding age, education level, and place of residence (city or village). The period of alcohol use initiation was divided into three groups: under 15 years of age, 16–18 years of age, and over 18 years of age. The participants were also asked about alcohol consumption in the year prior to the interview. The type of alcohol was converted into amounts: consuming more than 12 g/day was considered risky. Data on height and body weight were used to calculate current BMI according to the following formula: body mass (kg) divided by height (m) squared. Participants in the study were asked about their physical activity: daily physical activity (a minimum of 30 min of exercise or a walk of 10,000 steps) was classified as high, physical activity 2–3 times per week was classified as moderate, and physical activity once a month or less frequently was classified as lack of physical activity. Smoking habits (i.e., current and former smoking, total years of smoking, and the number of cigarettes smoked per day) were also recorded. Family history of breast cancer, gynaecological history (the occurrence of menarche, age at menopause, and use of hormone replacement therapy (HRT)), age at first pregnancy, miscarriage/abortion, number of children, and length of breastfeeding were recorded during the interview. Questions regarding mononucleosis history and exposure to ionising radiation during early childhood were also included in the questionnaire.

### 2.3. Statistical Analysis

Data from the questionnaire were input into Statistica 13. The Student’s t-test was used for independent samples to evaluate the mean differences of normally distributed variables (i.e., BMI) between cases and controls; in the case of skewed variables, the tested hypothesis was evaluated using the non-parametric Mann–Whitney U test. The univariate analysis was used to evaluate the OR and 95% CI and to estimate the relationship between individual risk factors and breast cancer. Multivariate logistic regression was applied to evaluate the association between breast cancer and education, alcohol initiation, breastfeeding, place of residence, and BMI. The analysis was adjusted for potential confounding factors, such as age, smoking (ever), physical activity (none, moderate, high), family history of breast cancer, age at menarche, age at first pregnancy, radiation in childhood, childhood illnesses, inter alia mononucleosis, and use of HRT. The variables were selected using a backward stepwise regression. Based on the ultimate model, the significance level was set at 0.05. The results of the logistic regression analysis were presented as odds ratios (ORs) with 95% confidence intervals (95% CIs).

## 3. Results

The basic characteristics of women who participated in the study are presented in Table 1.

Cases and controls were of similar age (mean age: 58 ± 6 years old). Cases started consuming alcohol earlier in life as compared to controls (17% vs. 9%; *p* = 0.05), less frequently reported breastfeeding (>3 months; 52% vs. 61%, *p* < 0.001), and were more often obese as compared to healthy women (34% vs. 20%; *p* = 0.002). Moreover, cases frequently only had primary education (66.67% vs. 33.33%; *p* = 0.006) and lived in rural areas (25% vs. 15%; *p* = 0.01). There were no statistically significant differences regarding smoking, drinking over 12 g alcohol per day (in the 1 year preceding the interview), physical activity, and menopausal status between the groups. Women in both groups were also characterised by similar family histories (the occurrence of breast cancer in the family, as well as pathologies predisposing them to the disease, i.e., hereditary breast cancer, site-specific- breast and ovarian cancer syndrome, Li-Fraumeni syndrome, Lynch II syndrome, Cowden’s disease, Peutz-Jeghers syndrome, ataxia-telangiectasia, and Klinefelter syndrome). The higher use of HRT among cases as compared to controls (OR = 1.58, 95% CI: 0.98–2.52, *p* = 0.06) is also worth noting because it could turn out to be significant in a larger population. The univariate analysis (Table 2) did not show a relationship between breast cancer and the following risk factors: history of miscarriage or abortion, age at menarche, smoking, drinking over 12 g alcohol per day, exposure to radiation, physical activity, and history of mononucleosis.

Table 3 shows the multivariate analysis results for early alcohol initiation, obesity, not breastfeeding, residence in a rural area, and primary and vocational education. The analysis was adjusted for potential confounding factors such as: age, a history of smoking, drinking alcohol over 12 g/d, physical activity, a family history of breast cancer incidence, diseases predisposing one to breast cancer, age at menarche, age at first pregnancy, childhood diseases, the use of HRT, and menopausal status.

## 4. Discussion

In our study, we demonstrated that the following risk factors are associated with an increased risk of cancer: early alcohol initiation, breastfeeding for less than 3 months, obesity, living in a rural area, and a lower level of education.

The above study showed a positive relationship between alcohol consumption in early youth (≤15 years) and an increased risk of breast cancer. These findings are noteworthy because the most well-known risk factors for breast cancer seem to play a role in early life during breast development and puberty [28]. Ying Liu et al. reported that for every drink/day consumed before the first pregnancy, the risk of breast cancer increased [29]. In a meta-analysis of 53 studies, researchers estimated that adult women who consumed alcohol daily are at an increased risk compared with women who abstain from alcohol [30]. Other studies have also shown a positive relationship between alcohol consumption and an increased risk of breast cancer [31,32,33,34]. A Japanese study that shed more light on the effects of alcohol consumption on breast cancer found that the risk was greater only in premenopausal women [35]. Unfortunately, despite the fact that many studies on alcohol consumption among adult women have been conducted in the context of breast cancer, little research has focused on the effects of alcohol consumption in childhood and adolescence. It is precisely this gap that must be filled.

The influence of alcohol on the risk of breast cancer is most often explained by the fact that alcohol affects the levels of circulating sex hormones, where high levels of oestrogen in the blood are associated with an elevated risk of breast cancer. Alcohol increases the concentration of oestrogen in the blood, rendering the oestrogen concentration particularly high in the middle of the menstrual cycle (27–38% higher than in abstainers) when the oestrogen levels are already high by nature; hence the conclusion that alcohol can contribute to the processes of neoplasia more strongly in adolescence than in adulthood [36]. Unfortunately, only a small percentage of the population is aware that alcohol has a different effect on the female body than on the male body, with the risk of negative consequences being several times greater in women. In Poland, alcohol consumption is growing steadily, along with a simultaneous lack of support for alcohol use disorder treatment. Statistics show that 85% of all teenagers regularly consume alcohol [37]. Moreover, in Europe, 30% of 15-year-old girls reported that they first began consuming alcohol at the age of 13 or younger [38]. Today, additional public attention should be given to identifying and preventing underage girls from drinking alcohol because they will be at an increased risk of breast cancer over the next 50 years. Hence, future studies should examine drinking patterns over the lifetime in relation to the risk of breast cancer. In addition, little is known about the lifestyle components that could modify the adverse effects of alcohol consumption on breast cancer development [29].

In congruence with previous epidemiological studies, our study showed that the lack of or only a short period of breastfeeding, obesity, and residing in a rural area are associated with an increased risk of breast cancer. A meta-analysis of cohort and case-control studies has shown that breastfeeding women, regardless of ethnic and cultural conditions, have a lower risk of breast cancer compared to women who never breastfed [39]. Other studies support this hypothesis [4,40]. Ambrosone et al. showed an increased incidence rate of triple-negative breast cancer among African-American women, with a small percentage of those women being breastfeeding women [41,42]. Breastfeeding has been classified by the World Cancer Research Fund (WCRF) as a protective factor against breast cancer in both pre- and post-menopausal women [43]. The protective mechanism of breastfeeding is believed to result from the reduction of the number of menstrual cycles, which is associated with reduced exposure to endogenous oestrogens, which may increase the differentiation of duct cells, making them less susceptible to carcinogens [44,45]. Breastfeeding may also facilitate the excretion of carcinogens via milk ducts [45,46].

Our study also confirms the importance of obesity as a risk factor for breast cancer. The WCRF recognised obesity in adulthood as a factor increasing the risk of post-menopausal breast cancer [47]. Moreover, the results of a study by Neuhouser et al. confirmed this positive relationship [48]. In a meta-analysis, Renehan et al. estimated that any increase in BMI after menopause is associated with an increase in the risk of breast cancer [49]. Other studies have also shown a positive relationship between obesity and breast cancer [50,51]. Obesity is a factor affecting oestrogen levels after menopause; in adipose tissue, androstenedione aromatisation creates oestrone, a weak oestrogen, while excess fat (an organ with endocrine secretion) causes an increase in oestrogen levels, which then contributes to an increased risk of breast cancer [52,53,54]. This study did not include a BMI from the adolescence period. Available analyses show an inverse relationship between being overweight or obese before the first pregnancy and the risk of premenopausal breast cancer. The study was retrospective—it was difficult for the respondents to determine their own body weight and height from the period 40 years ago. Therefore, a prospective study should be planned to investigate the effect of BMI and alcohol in adolescence on the risk of breast cancer.

The results of this study also indicate that residents of rural areas have a significantly higher risk of breast cancer than residents of urban areas. A similar trend has been observed in England [55]. In Poland, the reasons for this can be found in the contamination of the environment with DDT and its metabolites. After the Second World War, the inhabitants of the Polish countryside largely ate potatoes. The Colorado beetle, which appeared in the early 1950s, initiated the massive use of organochlorine pesticides, which lasted until the end of the 1970s when these measures were banned. This is the period of foetal life, birth, and adolescence for the study participants. Taking into account the results of the Howe and McLaughlin studies, it is early in life when exposure to environmental factors (radiation) may increase the risk of breast cancer [56]. Atomic bomb survivors < 20 years of age had the greatest excess risk of breast cancer [57]. These women have mostly been <20 years of age as DDT use rose to its peak. They were exposed to high doses of DDT both in the prenatal period (through the mother’s body), as infants (through milk), and later) due to contaminated food, water, or air) [58]. Research suggested that exposure to DDT during childhood and early adolescence (younger than 14 years) was associated with a five-fold increase in the risk of developing breast cancer before age 50 (21). There are few studies available on the impact of the use of organochlorine pesticides, such as DDT, on the risk of breast cancer—especially in adolescence. Because we have not used these compounds for over 50 years, we are slowly forgetting about them, but they will be us and the next generations for many years to come.

### Limitation

As in all case-control studies, the major limitation of this study was recall bias. However, an effort was made to minimise this limitation by choosing newly diagnosed consecutive patients and recruiting the necessary participants in a short period of time. Another limitation is that the study was performed only among Caucasian women who have a high rate of breast cancer. Therefore, the results of the study cannot be generalised to the entire population. Furthermore, the study did not include a question about the use of hormonal contraception, which may be an additional limitation. Furthermore, bodyweight was not queried prior to diagnosis, which may be an additional limitation of the study. However, it was assumed that the study would include patients with a diagnosis not older than 3 months, which should minimise such error.

## 5. Conclusions

In conclusion, we found that early alcohol use initiation, breastfeeding for less than 3 months, obesity, living in a rural area, and a lower education level were associated with an increased risk of breast cancer in our study sample. Furthermore, the results of our study indicate the need to create breast cancer prevention programs aimed at girls and young women to ensure awareness regarding alcohol consumption at an early age, before the first pregnancy, as a risk factor for breast cancer.

## Figures and Tables

**Table 1 cancers-13-03925-t001:** The characteristics of cases and controls.

	Cases	Controls	
Age (years) mean ± SD	59 ± 6	58 ± 6	*p*
Characteristics	*n* (%)	*n* (%)	
Educational level	*n* (% of level)		
Primary	22 (66.67)	11 (33.33)	
Vocational	40 (68.97)	18 (31.03)	0.006
High school	71 (41.28)	101 (58.72)	
University degree	68 (48.92)	71 (51.08)	
Physical activity	n (% of all)		
No	102 (51)	110 (55)	
Moderate	63 (31)	50 (25)	0.7
High	36 (18)	41 (20)	
Smoking			
Never	85 (43)	90 (45)	
Former	86 (43)	69 (34)	0.3
Current	30 (14)	42 (21)	
Age of alcohol initiation (year)			
≤15	35 (17)	19 (9)	0.05
16–18	74 (37)	79 (39)	
>18	92 (46)	103 (52)	
Age at menarche (year)			
≤12	39 (19)	40 (20)	0.9
>12	162 (81)	161 (80)	
Mononucleosis in childhood			
Yes	3 (1.5)	3 (1.5)	
No	171 (85)	178 (88.5)	0.5
I do not know	27 (13.5)	20 (10)	
Number of live births			
0	34 (17)	25 (12)	
1	44 (22)	53 (26)	
2	73 (36)	77 (38)	0,9
3	29 (14)	28 (14)	
≥4	21 (11)	18 (10)	
Breastfeeding (>3 months)			
Yes	104 (52)	123 (61)	<0.001
No	97 (48)	78 (39)	
Hormone replacement therapy *			
Yes	54 (27)	38 (19)	0.06
No	147 (73)	163 (81)	
Age of menopause:			
Premenopausal	41 (20)	52 (26)	0.9
≤55	98 (49)	98 (49)	
>55	62 (31)	51 (25)	
Family history of breast cancer:			
Yes	52 (26)	45 (22)	
No	149 (74)	156 (78)	0.4
Family history of breast cancer < 55-year-old			
Present			
Absent	30 (15)	21 (10)	0.2
	171 (85)	180 (80)	
Residence area:			
Urban	151 (75)	171 (85)	0.01
Rural	50 (25)	30 (15)	
BMI **			
≤29.99	133 (66)	161 (80)	0.002
≥30	68 (34)	40 (20)	
Alcohol > 12g/day			
Yes	15 (3.76)	10 (2.49)	0.37
No	186 (46.27)	191 (47.51)	

* HRT—Hormone replacement therapy; ** BMI—Body mass index.

**Table 2 cancers-13-03925-t002:** Results of univariate logistic regression.

	OR *	95% CI **	*p*
Rural residence area	1.9	1.15–3.14	0.01
Lower educational level	2.73	1.66–4.47	0.0001
Lack of physical activity	1.17	0.71–1.93	0.5
Smoking	1.1	0.75–1.64	0.6
Exposure to radiation	1.4	0.77–2.57	0.3
History of mononucleosis	1.33	0.76–2.35	0.3
Age at menarche > 12 y	1.03	0.63–1.69	0.9
History of miscarriage or abortion	1.12	0.7–1.8	0.63
Menopause	0.92	0.66–1.45	0.98
HRT ***	1.58	0.98–2.52	0.06
Family history of breast cancer	1.2	0.76–1.9	0.4
Diseases predisposing one to breast cancer	1.45	0.54–3.9	0.45
BMI **** ≥ 30	2.06	1.3–3.24	0.018
Alcohol initiation > 15	0.51	0.27–0.94	0.03
Alcohol > 12 g/day	1.54	0.67–3.52	0.3
Breastfeeding < 3 months	2.44	1.63–3.63	<0.001
Educational level—secondary	1		
Educational level—primary	3.16	1.68–5.96	0.0004
Educational level—vocational	2.84	1.3–6.23	0.009
Educational level—high	1.3	0.85–2.1	0.2

* OR—Odds ratio; ** CI—Confidence intervals; *** HRT—Hormone replacement therapy; **** BMI—Body mass index.

**Table 3 cancers-13-03925-t003:** Results of multivariate logistic regression.

	OR *	95% CI **	*p*
BMI *** ≥ 30	1.9	1.16–3.04	0.009
Alcohol initiation ≤ 15 y	1.98	1.06–3.69	0.03
Breastfeeding below 3 months	2.31	1.52–3.50	<0.001
Rural residence	1.7	1.05–2.9	0.05
Primary and vocational education	2.5	1.49–4.20	<0.001

* OR—Odds ratio; ** CI—Confidence intervals; *** BMI—Body mass index.

## Data Availability

The data presented in this study are available on request from the corresponding author. The data are not publicly available due to privacy.

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
