# Peer review of "Early Alcohol Use Initiation, Obesity, Not Breastfeeding, and Residence in a Rural Area as Risk Factors for Breast Cancer: A Case-Control Study"

_cancers, 2021, doi:10.3390/cancers13163925_

Round 1

Reviewer 1 Report

The Authors didn't answer to my questions correctly. The design of the study is unclear and the conclusions not supported by the results

Author Response

We stand by our opinion that our analysis method was correct and it does not require any improvement. We appreciate comments from the Reviewer 1 but we are convinced that the further changes in our manuscript in accordance with His recommendation are not possible as the suggested data were not collected.

We presented a classical statistical method for data analysis of the case-control study, which is logistic regression. If the Reviewer would be more specific on what method would be more appropriate we could consider it.

We did not collect data on diabetes and metabolic syndrome therefore it is not possible to include it into the model. Nevertheless, including the diabetes or metabolic syndrome into the model would not add any additional significant explanation. We will include this information in future research when it is collected. The same refers to circumference waist. Findings from several studies confirmed that a larger waist circumference is associated with higher risk of postmenopausal breast cancer, but not beyond its contribution to BMI (Gaudet MM, Carter BD, Patel AV, Teras LR, Jacobs EJ, Gapstur SM. Waist circumference, body mass index, and postmenopausal breast cancer incidence in the Cancer Prevention Study-II Nutrition Cohort. Cancer Causes Control. 2014 Jun;25(6):737-45. doi: 10.1007/s10552-014-0376-4. Epub 2014 Apr 9. PMID: 24715420). Therefore we had BMI in our analysis without circumference waist which seemed to be redundant if included into analysis in our study.

We used stepwise regression to selects explanatory variables for multiple regression models based on their statistical significance. Stepwise regression method automatically skip the variables, which were not significant.

Reviewer 2 Report

Thank you for addressing my earlier comments. I have no further comments. 

Author Response

NA

This manuscript is a resubmission of an earlier submission. The following is a list of the peer review reports and author responses from that submission.

Round 1

Reviewer 1 Report

This is an anecdotal and incomplete paper which describes the predictive value of obesity and early alcohol use initiation on breast cancer. First of all the statistical analysis is not adequate and well described. Furthermore the Authors should add a chapter which analyzes circumference waist, diabetes and metabolic syndrome as different risk factors for breast cancer. Finally, the number of subjects studied is too small to make conclusive remarks. So, the conclusions are not supported by the results and the predictive value of obesity and early alcohol use initiation is weak

Author Response

Dear Reviewer.

Thank you very much for your valuable comments and suggestions.

  1. We have completed the description of the statistical methods used.
  2. A sample size of 402 women (201 cases and 201 controls) was estimated to be necessary to observe a significant association between early alcohol use initiation and the risk of breast cancer. The sample size calculation was performed using a standard formula- taking into consideration an estimated prevalence of early alcohol use initiation among controls of 29% , an odds ratio of 1.8, a case to control ratio of 1:1, a power of 80%, and a significance level of 5%. The sample size analysis was per-formed in Stata (StataCorp. 2013. Stata Statistical Software: Release 15.1.; StataCorp LP, College Station, TX, USA) using commands that accounted for the matched case‐control nature of the study design.
  3. We are aware of the limitations of our research. We did not include diabetes and metabolic syndrome in the questionnaire. This is a very valuable comment that we will use in planning further research. Incidences of breast cancer, type 2 diabetes, and metabolic syndrome have increased over the past decades with the obesity epidemic, especially in industrialized countries. We made the assumption (perhaps this is our mistake) that breast cancer, metabolic syndrome and diabetes share many risk factors, such as obesity and a sedentary lifestyle, and potentially dietary factors such as intake of saturated fat and refined carbohydrates, although their role is less clear for breast cancer etiology than for diabetes. Hence, the observed association between diabetes and breast cancer risk may be partly due to the clustering of the 2 disorders as a consequence of shared risk factors. Up to 16% of patients with breast cancer who are older than 65 years also have diabetes mellitus.Thus, the incidence of both breast cancer and type 2 diabetes is high in elderly people and both share a common risk factor—obesity. Otherwise- in clinical practice, hyperinsulinemia is most often the result of increasing insulin resistance, which is a consequence of overweight and obesity. Renehan et al. In 2008 presented a meta-analysis of data from 141 studies, showing that obesity is associated with an increased risk of cancer incidence in both sexes.

Thank you very much again for reviewing our article.

Reviewer 2 Report

Thank you for the opportunity to review this manuscript. I have some comments below. 

  1. Simple summary. (a) Second sentence: "The aim of the study was to evaluate risk factors" in terms of what? More information is required.  (b) Why should prevention programs be aimed at primary school? 
  2. Introduction, fourth paragraph, lines 75-78. I would argue that these barriers do not increase the incidence of breast cancer, but rather lead to later detection and diagnosis (and so may increase mortality). Could an alternative explanation for a higher rate in rural areas be related to lifestyle factors perhaps? 
  3. What information on alcohol consumption was collected? This is not mentioned in the Methods. Was the effect of alcohol consumption overall investigated, given that alcohol consumption has been associated with an increased risk of breast cancer?  
  4. At what age was BMI measured? Current BMI may not necessarily represent BMI during the period of risk (particularly for cases), and this should be discussed. 
  5. Table 3. What other variables were entered into this regression (i.e. what variables were controlled for?). 
  6. Discussion, page 8, line 210-212. You mention the relationship between alcohol and obesity; however, your analysis presumably controlled for both such that early alcohol initiation was uniquely associated with breast cancer risk. Is that correct? This could be explained more clearly. 
  7. Likewise, were rurality and education entered into the model separately, and were they uniquely associated with breast cancer risk? If so, the discussion on paged 8-9, lines 256-261 may need some further explanation - the increased risk among rural residents cannot be solely attributed to their lower education level.
  8. There are English language and grammatical errors throughout the manuscript; some English editing would be beneficial. 

Author Response

Dear Reviewer.

Thank you very much for your valuable comments and suggestions.

  1. Simple summary. (a) Second sentence: "The aim of the study was to evaluate risk factors" in terms of what? More information is required.  (b) Why should prevention programs be aimed at primary school? 

We have made the recommended changes to the simple summary section.

  1. Introduction, fourth paragraph, lines 75-78. I would argue that these barriers do not increase the incidence of breast cancer, but rather lead to later detection and diagnosis (and so may increase mortality). Could an alternative explanation for a higher rate in rural areas be related to lifestyle factors perhaps?

This is a very valuable comment. Indeed, we cannot say that barriers related to education or the lack of access to a specialist clinic can increase the risk of developing the disease. We analyzed historical data and based on them we hypothesized that perhaps the exposure of rural women to DDT could increase the risk of developing the disease in this group. It is a pesticide that we have not used in Poland for over half a century, but we still find it in nature and food. As a student, I determined DDT in breast milk and in all samples that I tested, regardless of the lactation period, the amounts of DDT exceeded the norm.

  1. What information on alcohol consumption was collected? This is not mentioned in the Methods. Was the effect of alcohol consumption overall investigated, given that alcohol consumption has been associated with an increased risk of breast cancer?

We have collected information on alcohol use initiation (three age groups: under 15, up to 18 and over 18) and the amount of alcohol currently drunk (in the year preceding the study). Consuming more than 12g / day has been identified as risky.

  1. At what age was BMI measured? Current BMI may not necessarily represent BMI during the period of risk (particularly for cases), and this should be discussed.

Unfortunately, we did not obtain information on body weight and height in adolescence or in the period preceding the diagnosis - we introduced the above issue to the limitations of our study.

  1. Table 3. What other variables were entered into this regression (i.e. what variables were controlled for?).

We supplemented the information in the text with the variaboes that were taken into account in the regression.

  1. Discussion, page 8, line 210-212. You mention the relationship between alcohol and obesity; however, your analysis presumably controlled for both such that early alcohol initiation was uniquely associated with breast cancer risk. Is that correct? This could be explained more clearly.

We removed this suggestion from the discussion as it was not related to our analysis.

  1. Likewise, were rurality and education entered into the model separately, and were they uniquely associated with breast cancer risk? If so, the discussion on paged 8-9, lines 256-261 may need some further explanation - the increased risk among rural residents cannot be solely attributed to their lower education level.

We have made a correction in the discussion. Both factors were introduced together into the model. Thank you for your help - of course we agree that we cannot say that lower education increases the risk of breast cancer - perhaps it has an influence on the knowledge of risk factors and indirectly affects the incidence, but it has no direct relationship.

  1. There are English language and grammatical errors throughout the manuscript; some English editing would be beneficial.

The article has been English edited.

Thank you very much  for your valuable and helpful review of our manuscript.

Round 2

Reviewer 1 Report

The Authors didn't use a correct statystical analysis to demonstrate their results. They better explained the Methods, but the description is not sufficient. In addition, they didn't take into account circumference waist, diabetes and metabolic syndrome as independent risk factors for breast cancer and this is a serious problem. Finally, i suggest to enlarge the population study.

Reviewer 2 Report

Thank you for your response to my previous comments. I have a few further comments. 

Introduction, paragraph 4. While the presence of DDT is an alternative explanation for rurality being associated with higher breast cancer incidence, this is not assessed in the current study. I feel that this section could be shortened to a sentence or two explaining that a potential reason for the link between rurality and BC incidence is the presence of pesticides (i.e. the first sentence added here), with the rest of the paragraph being omitted as I feel it provides too much detail which is then not followed up in the manuscript itself. You could also note that other lifestyle factors (e.g. alcohol use, obesity) may be linked to rural residence (if that is the case in Poland as in other countries). 

Table 3. Given that there were no differences between groups in terms of age, smoking, alcohol consumption, etc, can a reason be given as to why these variables were controlled for in the regression analysis?